# Importance of the PD-1/PD-L1 Axis for Malignant Transformation and Risk Assessment of Oral Leukoplakia

**DOI:** 10.3390/biomedicines9020194

**Published:** 2021-02-16

**Authors:** Jutta Ries, Abbas Agaimy, Falk Wehrhan, Christoph Baran, Stella Bolze, Eva Danzer, Silke Frey, Jonathan Jantsch, Tobias Möst, Maike Büttner-Herold, Claudia Wickenhauser, Marco Kesting, Manuel Weber

**Affiliations:** 1Department of Oral and Maxillofacial Surgery, Friedrich-Alexander University Erlangen-Nürnberg (FAU), 91054 Erlangen, Germany; jutta.ries@uk-erlangen.de (J.R.); falk.wehrhan@outlook.de (F.W.); christoph.baran@uk-erlangen.de (C.B.); cipst_09@yahoo.com (S.B.); evadanzer@yahoo.de (E.D.); tobias.moest@uk-erlangen.de (T.M.); marco.kesting@uk-erlangen.de (M.K.); 2Institute of Pathology, Friedrich-Alexander University Erlangen-Nürnberg (FAU), 91054 Erlangen, Germany; abbas.agaimy@uk-erlangen.de; 3Private Office for Maxillofacial Surgery, 91781 Weißenburg, Germany; 4Department of Internal Medicine 3—Rheumatology and Immunology, Friedrich-Alexander University (FAU) Erlangen-Nürnberg and Universitätsklinikum Erlangen, 91054 Erlangen, Germany; silke.frey@uk-erlangen.de; 5Institute of Clinical Microbiology and Hygiene, University Hospital of Regensburg and University of Regensburg, 93053 Regensburg, Germany; Jonathan.Jantsch@klinik.uni-regensburg.de; 6Department of Nephropathology, Institute of Pathology, Friedrich-Alexander University Erlangen-Nürnberg (FAU), 91054 Erlangen, Germany; maike.buettner@uk-erlangen.de; 7Institute of Pathology, Halle (Saale) University Hospital, Martin-Luther University Halle-Wittenberg (MLU), 06108 Halle (Saale), Germany; claudia.wickenhauser@uk-halle.de

**Keywords:** immune checkpoints, PD-1, PD-L1, oral leukoplakia, OSCC, malignant transformation

## Abstract

Background: The programmed cell death ligand 1/programmed cell death receptor 1 (PD-L1/PD-1) Immune Checkpoint is an important modulator of the immune response. Overexpression of the receptor and its ligands is involved in immunosuppression and the failure of an immune response against tumor cells. PD-1/PD-L1 overexpression in oral squamous cell carcinoma (OSCC) compared to healthy oral mucosa (NOM) has already been demonstrated. However, little is known about its expression in oral precancerous lesions like oral leukoplakia (OLP). The aim of the study was to investigate whether an increased expression of PD-1/PD-L1 already exists in OLP and whether it is associated with malignant transformation. Material and Methods: PD-1 and PD-L1 expression was immunohistologically analyzed separately in the epithelium (E) and the subepithelium (S) of OLP that had undergone malignant transformation within 5 years (T-OLP), in OLP without malignant transformation (N-OLP), in corresponding OSCC and in NOM. Additionally, RT-qPCR analysis for PD-L1 expression was done in the entire tissues. Additionally, the association between overexpression and malignant transformation, dysplasia and inflammation were examined. Results: Compared to N-OLP, there were increased levels of PD-1 protein in the epithelial and subepithelial layers of T-OLP (p_E_ = 0.001; p_S_ = 0.005). There was no significant difference in PD-L1 mRNA expression between T-OLP and N-OLP (*p* = 0.128), but the fold-change increase between these groups was significant (Relative Quantification (RQ) = 3.1). In contrast to N-OLP, the PD-L1 protein levels were significantly increased in the epithelial layers of T-OLP (*p* = 0.007), but not in its subepithelial layers (*p* = 0.25). Importantly, increased PD-L1 levels were significantly associated to malignant transformation within 5 years. Conclusion: Increased levels of PD-1 and PD-L1 are related to malignant transformation in OLP and may represent a promising prognostic indicator to determine the risk of malignant progression of OLP. Increased PD-L1 levels might establish an immunosuppressive microenvironment, which could favor immune escape and thereby contribute to malignant transformation. Hence, checkpoint inhibitors could counteract tumor development in OLP and may serve as efficient therapeutic strategy in patients with high-risk precancerous lesions.

## 1. Introduction

In oral cancer, the programmed cell death ligand 1/programmed cell death receptor 1 (PD-L1/PD-1) signaling pathway represents an important immune checkpoint, which limits immune reactions and contributes to the tumor immune-escape [1]. Inhibitors of the PD-1 receptor are currently the only clinically approved checkpoint inhibitors in OSCC patients. PD-1 inhibitors are used in advanced stage OSCC with a missing curative surgical or radio-oncological treatment option. In some of these patients PD-1 blocking leads to long lasting responses [2]. Recently, the clinical use of checkpoint inhibitors is shifting towards early stages of OSCC treatment with the first results of successful neoadjuvant PD-1 inhibition being published [3,4] and currently evaluated in a large prospective study (KEYNOTE-689 study; NCT03765918).

The PD-1 receptor and its ligands PD-L1 and PD-L2 are important modulators of the immune system. The expression of PD-L1 on normal tissues is limited. However, numerous tumor cells overexpress PD-L1 as a strategy to evade immune responses. Thus, the expression of PD-L1 on tumor cells may play an important role in suppressing T cell immune activity and may help malignant cells to escape from the immune system [5]. We demonstrated earlier that PD-L1 and PD-1 expression in OSCC was significantly higher compared to NOM [6,7]. However, association of intratumoral PD-L1 expression with malignancy and prognosis was not detected [7]. This is in accordance with a recent meta-analysis showing no significant association with survival [8]. Nevertheless, there was evidence for an association of PD-L1 with some staging parameters [8]. Additionally, we found that an increased expression of PD-L1 in the peripheral blood of OSCC patients was associated with lymph node metastases and poor prognosis [7,9]. This indicates that the local as well as the systemic immune environment contribute to OSCC progression [7,10,11]. 

In the oral carcinogenesis most OSCC arise on the basis of potentially malignant oral mucosa lesions—oral leukoplakia (OLP) [12]. These are white patches that can easily be detected clinically. However, not all OLP have the potential of malignant transformation [12,13]. The gold standard is the histomorphologic assessment of dysplasia in OLP incision biopsies [14,15,16]. However, this method sometimes failed to assess the potential risk of malignant transformation of the OLP. Currently, there is no method available to reliably judge the risk of OLP malignant transformation. Hence, better predictive or aiding marker is urgently needed. Today, there are emerging molecular biomarkers, but none of them were reliable enough to be included in routine diagnostics. Nevertheless among these markers supporting the diagnosis, some immunohistochemical and molecular biomarkers are proposed to have prognostic potential [17,18,19,20,21,22]. Additionally, at the moment, there are ongoing efforts to use additional cellular and immunologic markers to better predict the risk of OLP malignant transformation (PREDICT-OLP study; NCT03975322).

There is strong evidence that immunologic alterations do contribute to the progression of OLP and occur prior to malignant transformation [23]. OLP with a malignant transformation within a time interval of five years display an increased rate of immunosuppressive, “M2-like” macrophages [23]. 

The immune system plays an important role recognizing tumor- and tumor precursor cells. However, due to immunoediting mechanisms like checkpoints cancer cells are able to escape immune surveillance and establish clinically apparent cancer diseases [24,25]. 

PD-1/PD-L1 Immune Checkpoint overexpression is involved in immunosuppression and the failure of an immune response against tumor cells by inhibition of T cell effectiveness. Hence, this mechanism contributes to tumor immune escape and expression analysis of these immune modulators may predict the risk of transformation. 

There is limited information on PD-L1 and PD-1 expression between OLP with malignant transformation (T-OLP) and OLP without malignant transformation (N-OLP). An immunohistochemical analysis showed an increased PD-L1 expression in T-OLP compared to N-OLP [26]. However, only eight cases with malignant transformation were included in this report [26]. 

The study aims to analyze the expression of PD-L1 and PD-1 in healthy normal oral mucosa (NOM), OSCC, T-OLP and N-OLP using RT-PCR and immunohistochemistry and to correlate the PD-1/PD-L1 expression values to malignant transformation and infiltration of inflammatory cells in OLP. 

## 2. Materials and Methods

### 2.1. Patients Collective

For this retrospective bicentric study, paraffin-embedded tissue samples were examined. They were taken from historical patient collectives of the universities of Erlangen and Halle (Saale) between 1997 and 2015. A sampling technique known as consecutive sampling or total enumerative sampling was used to obtain the samples in both centers. Hence, every subject who met the inclusion criteria of one of the four groups was included in the study until the required sample size was reached. Tissue samples were collected with the consent of the patients and approved by the ethics committee (application number 3962; date: 16 April 2009, prolongation: 1 December 2010). The specimens were histopathologically evaluated by three pathologists and divided into four groups. Group 1 included tissue samples from patients with oral leukoplakia, which developed into squamous cell carcinoma within 5 years, called T-OLP. The associated samples of oral squamous cell carcinoma were assigned to group 3, called OSCC. Group 2 included oral leukoplakia without transformation within 5 years, called N-OLP. All patients suffering from an OLP were followed up for at least 5 years. The interval between the first diagnosis of OLP and malignant transformation, also called disease free survival (DFS), was determined. Group 4 included samples of NOM. Grades of dysplasia of all OLP (A-OLP) were histopathologically classified into dysplasia grades in accordance with the World Health Organization (WHO) classification of tumors of the Head and Neck 2017 [27]. According to the consensus between two pathologists (out of three) the OLP were classified as follows: D0 for no dysplasia, D1 for mild, D2 for moderate and D3 for severe dysplasia/carcinoma in situ (CIS). Additionally, the OLP were grouped into low risk (low-grade D0/D1) and high-risk lesions (high-grade D2/D3). Initial management consisted of surgical excision, CO_2_ laser vaporization or observation only. The OLP patients were treated according to the severity of dysplasia as recommended in the guidelines for management of OLP [28]. Thus, leukoplakia with no or mild dysplasia were controlled at regular intervals of 6 months. Moderate and high dysplastic OLP were surgically excised or laser removed as completely as possible.

Corresponding OSCC were classified according to tumor size (grouped into T1&2 and T3&4), the state of the lymph node joined together as N0 and N+ to indicate the absence (N0) or presence (N+) of metastases, their differentiation (well-differentiated (G1), moderately (G2) and poorly differentiated (G3)) and their clinical stage (gathered in early (stage I & II) and late (stage III & IV) stages). Lastly, all samples were histologically categorized according to the inflammatory infiltration. They were classified as “none”, “mild”, “moderate” and “severe” based on the sections at 10× magnification according to infiltration density by inflammatory cells in the subepithelium of the affected tissue. Samples that did not present any inflammatory infiltration were categorized as having “no inflammation”. Mild inflammation describes a few, scattered inflammatory cells, moderate pronounces individual inflammatory foci or scattered inflammatory cells, while severe describes several, confluent inflammatory foci or many scattered inflammatory cells in the affected epithelium. Representative examples are shown in Appendix A. Furthermore, the samples were divided into two groups: no to mild inflammation and moderate to severe inflammation.

The demographic, clinical and histopathological characteristics of patients are summarized in Table 1 and Table 2.

Due to the limited amount of material, not all samples that were immunohistologically examined could be included in the PCR analyses. Therefore, the number of samples and the demographic data of the patient collectives studied by the two methods differ from each other. However, for general statistical analyses concerning localization and inflammation and the association of these parameters to malignant transformation, and the disease free survival (DFS 0 time up to malignant transformation), Kaplan–Meier graph and log rank test for survival were carried out with the smaller collective examined by the PCR method.

A total of 162 formalin-fixed paraffin-embedded (FFPE) tissue samples were analyzed for PD-L1 expression by real time (RT) PCR. The T-OLP group included 32, the N-OLP group 50 samples. The control groups encompassed 38 (OSCC) and 42 samples (NOM) (Table 1). All four groups were examined immunohistochemically for the expression of PD-1 and PD-L1. The total number of samples amounted to 108 for PD-1 and 164 for PD-L1 analysis (Table 2). 

### 2.2. Detection of PD-L1 Expression by Quantitative Real Time Reverse Transcriptase Polymerase Chain Reaction (RT-qPCR) 

The total RNA was isolated from formalin-fixed paraffin-embedded (FFPE) samples using the RNeasy FFPE^TM^ Kit (Qiagen, Hilden, Germany) according to the recommendations given by the manufacturer. The material tested contained at least 85% of the dysplastic or hyperplastic tissue from oral leukoplakia. The concentration and quality of the RNA was determined using the NanoDrop (PeqLab, Erlangen, Germany).

Subsequently, the isolated RNA was translated to cDNA using the Transcriptor High-fidelity cDNA Synthesis Kit according to the manufacturer’s instructions (Roche, Mannheim, Germany).

The cDNA was used to analyze the expression level through real-time quantitative PCR (RT-qPCR). Therefore, specific gene assays were selected from the TaqMan online database using the assay search tool on the Thermo Fischer website for the target genes PD-L1 (Hs00204257_m1) and the house keeping gene GAPDH (Hs02758991_g1). The TaqMan Fast Advanced Master Mix from Applied Biosystems (Life Technologies, Darmstadt, Germany) was utilized for amplification according to the protocol provided. The UNG incubation time was 2 min at 50 °C, the polymerase activation required 20 s at 95 °C, and the PCR (50 cycles) calls for 3 s for the denaturing process at 95 °C and 30 s for the annealing/extending step at 60 °C. GAPDH was used as the endogenous control and the tonsil was applied as a positive control. The samples were analyzed in duplicates. The data were collected and evaluated based on the specifications given by the manufacturer on the ABI Prism 7300 (Applied Biosystems, Darmstadt, Germany). The average CT-values obtained were used for further evaluations.

Samples that were included in the expression analysis show a clear amplification curve for the endogenous control, which a CT value <37. Samples showing a higher CT or no expression for GAPDH were removed from the study, as this indicates the poor quality or lack of quantity of the probe. Samples that received an acceptable CT value for the endogenous control but did not give a CT value for the target gene within the performed PCR cycles were declared as “Non-detects”. There the CT value was determined based on the maximum number of cycles fulfilled, which was 50 in this study [29]. The normalization of the CT values was performed by the ΔCT method using the house-keeping gene GAPDH as an internal control. The formula 2−ΔΔCT was used to calculate the relative alteration in expression rates between the two groups (RQ, FC).

### 2.3. Detection and Quantitative Immunohistochemical Analysis of PD-1 and PD-L1 Expression by Immunohistochemistry

Sections of 4 µm of the paraffin specimens fixed in formalin were made and then histopathological examination was carried out. The staining for the detection of PD-L1 was performed in the Institute of Pathology at the University Erlangen. After pretreatment in the solution Cell Conditioning 1 (CC1, Roche Diagnostics, Mannheim, Germany) for 50 min at 100 °C and incubation for 30 min at 36 °C, the samples were stained with the detection system OptiView BenchMarkUltra (Roche Diagnostics, Mannheim, Germany) using the antibody DAKO 22C3 (dilution 1:50 in antibody diluent, DAKO, Hamburg, Geremany).

The staining for detection of PD-1 was done in the Department of Oral and Maxillofacial Surgery at the University Hospital Erlangen. After pretreatment in a water bath at 100 °C for 20 min with Antigen Retrieval Buffer 4 (EDTA buffer, pH 9.0, Medac PMB4-125) followed by cooling at room temperature for a further 20 min, the samples were analyzed for PD-1 expression applying the Anti-PD1 antibody (ab137132, clone EPR4877/2, Abcam, dilution 1:500) and the DAKO Detection Kit K5001 (Dako, Hamburg, Germany) according to the manufacturer’s recommendations.

For both antigens membranous staining was defined as a positive result. Representative stains for all immune checkpoints examined are shown in Figure 1, Figure 2 and Figure 3. Quantitative analysis in the epithelial and the subepithelial compartment of the specimens was performed independently. Therefore, all samples were completely scanned and digitized using the method of “whole slide imaging” and the Pannoramic 250 Flash III Scanner. By the Pannoramic Viewer 1.15.2 software (3DHISTECH^®^, Budapest, Hungary) for each sample three epithelial and subepithelial image fields (Region of Interest, ROIs) were created. Care was taken to cover the stratifications of the epithelium. Subsequently, the pictures were exported into the TIF format by using the 3DHistech Slide Converter. Then the stains were quantitatively evaluated with the help of the program Biomas (MSAB, Erlangen, Germany). For this purpose, 100 cells of the three ROIs were counted in the epithelia and the positive stained cells were determined. Subepithelially, all cells and the positively stained cells of the image field were counted, so that at least 300 cells were considered. The ratio of positive cells to the total number of cells within the three ROIs was determined and the mean value was calculated from the triplets. This percentage values were used in the statistical expression analysis, now called the labeling index (∆LI).

### 2.4. Statistics

In qPCR analysis the PD-L1 relative gene expression between groups, represented as fold change (FC), were calculated using the ∆∆CT-method. A value greater than 2 is regarded as relevantly increased. The FC in immunohistological staining corresponded to the ratio of mean LI of the groups (∆LI1/∆LI2).

For evaluation of the results, the statistical software package SPSS 23 (SPSS Inc., Chicago, IL, USA) was used. It is based on the data collected from the PCR (ΔCT values) and the IHC (∆LI). Prior to the statistical analysis, the data were tested for their normal distribution by utilizing the Shapiro–Wilk test. In order to visualize the data box–whisker plots were utilized by displaying the median, interquartile range and minimum and maximum values of the gene expression in the different groups. Non-parametric test were used because the data are not normally distributed. Kruskal–Wallis test and Mann–Whitney U test (MWU) was done to determine whether the expressions between the groups differ significantly in the expression levels of the genes. A *p*-value ≤ 0.05 was considered statistically significant. The data were additionally examined using receiver operating characteristic (ROC) curves and the corresponding Area Under the Curve (AUC). A cut off points (COP) value was calculated. The COP allows one to derive a collective into two subgroups showing expression of PD-1/PD-L1, either above (underexpression, negative) or below (overexpression, positive) the COP. After this subclassification, the chi-square test (χ2-test) was used to explore whether the overexpression of PD-1/PD-L1 was associated with diagnosis, malignant transformation, dysplasia in OLP or TNM classification of OSCC and degree of tissue inflammation. A statistical significance was defined at a *p*-value ≤ 0.05. Lastly, the positive and negative predictive values were calculated. Additionally, in order to examine the relationship between the expression of the checkpoints and DFS of OLP a Kaplan–Meier graph is added and a log-rank test was made.

## 3. Results

### 3.1. Demographic, Clinical and Histomorphologic Characteristics of the Study Groups

The demographic, clinical and histopathological data of the bicentric patient collectives analyzed by different methods are summarized in Table 1 (for RT-qPCR analysis) and in Table 2 (for IHC analysis). In total 99 oral leukoplakia samples, 45 OSCC and 20 NOM were analyzed in the current study. Due to the limitations of the available material, the number of cases that were analyzed for expression of the markers by different methods, differed within the groups. All groups matched in gender (*p* > 0.05). The mean age of patients with T-OLP was marginally higher compared to patients with N-OLP. The samples of corresponding OSCC were obtained from the same patients as in T-OLP group. Subsequently, the mean age of the OSCC group was slightly higher than in T-OLP. The 20 control patients of NOM were significantly younger than OLP and OSCC patients (*p* < 0.001).

In the cohorts analyzed by different methods on 24% up to 29% of T-OLP in contrast to 2% of N-OLP were histomorphologically classified as “high-risk” lesions (D2/D3) (*p* = 0.005; Table 1 and Table 2). Histopathological and staging parameters of corresponding OSCC and inflammation grade of all tissues are given in Table 1 and Table 2.

### 3.2. Descriptive Statistics of PD-1 and PD-L1 Expression in T-OLP, N- OLP, OSCC and NOM

The mean level of the ∆CT values of the different groups varies (Table 3). NOM had the highest ∆CT average value (∆CT_PD-L1_ = 8.374). This means that in this group the average expression of PD-L1 was the lowest. T-OLP has the lowest ∆CT average value and hence the highest expression (∆CT_PD-L1_ = 4.864) followed by OSCC (∆CT_PD-L1_ = 5.759) and N-OLP (∆CT_PD-L1_ = 6.474). The expression of PD-L1 mRNA was 3-fold.

The IHC-determined subepithelial expression of PD-1 showed that the highest LI was given in the group of T-OLP (1.011), followed by NOM (0.66). A low identical expression was found for OSCC and N-OLP (0.2). Elevated mean values were also found in the epithelial compartment for T-OLP and NOM, while N-OLP and OSCC were lower and very similar. Sub- and epithelial expression of PD-1 was approximately 5 times higher in the T-OLP group than in the N-OLP group.

The highest epithelial expression of PD-L1 was found in the OSCC group (LI = 10.4), followed by T-OLP (6.1). With an LI of 0.6 epithelial expression of N-OLP was 10 times lower than that of T-OLP. The lowest expression was found in NOM (LI = 0.1). In the subepithelial compartment, expression rates varied marginally. The results are summarized in Table 3.

### 3.3. Comparison of Expression Rates between the Pathologically Altered Tissues and NOM

In order to check the association between expression levels and diagnosis the expression rates of PD-1 and PD-1L in NOM were compared to the OLP and OSCC group.

In RT-qPCR studies a statistically significant overexpression of the PD-L1-mRNA was observed in T-OLP (FC_PD-L1_ = 11.4, *p* = 0.0001), N-OLP (FC_PD-L1_ = 3.8, *p* = 0.0001) and OSCC (FC_PD-L1_ = 8.9, *p* = 0.0001) in comparison to NOM (Figure 4d, Table 4). The group of T-OLP had the highest average FC, followed by the OSCC group. By generating the ROC-curves, COPs could be calculated. If the samples were grouped into a positive (∆CT ≤ COP) or negative (∆CT ≥ COP) subgroup, there was a statistically significant difference in the distribution of positive and negative samples between OSCC (*p* = 0.0001) and NOM, and between T-OLP/N-OLP and NOM (*p* = 0.0001/*p* = 0.001) (Table 4).

Immunohistochemical studies showed that epithelial PD-L1 expression strongly altered in the tissues of T-OLP and OSCC compared to NOM. In T-OLP a significant 60-fold overexpression (*p* = 0.04) and in OSCC a 99.4-fold increase of PD-L1 was detected. No significant overexpression compared to NOM could be proven for N-OLP tissues (FC = 6.3; *p* = 0.70). The statistical relevance was confirmed by the AUC values (Table 4). Overexpression was additionally related to malignant transformation (*p* = 0.03) and malignancy (*p* = 0.003). Neither subepithelial PD-L nor PD-1 expression (epithelial/subepithelial) was significantly changed in the T-OLP group compared to NOM (Table 4). The subepithelial PD-L1 and PD-1 expression level (epithelial/subepithelial) was only slightly reduced in the N-OLP and OSCC groups compared to NOM, but the difference in expression rates between the groups was significant with the exception of subepithelial expression of PD-1 in OSCC (4).

### 3.4. Comparison of T-OLP and N-OLP—Association between Overexpression of PD-1 and PD-L1 and Malignant Transformation

In RT-qPCR analyses no statistically significant altered PD-L1 gene expression was observed between T-OLP and N-OLP (p_PD-L1_ = 0.128) (Figure 4d). As a result, a cut of point could not be determined to separate T-OLP and N-OLP. However, the average mRNA expression of PD-L1 was substantially higher in T-OLP compared to N-OLP (FC_PD-L1_ = 3.04). That indicates a tendency for evaluated PD-L1 expression in malignant transformation (Table 5).

The PD-L1 protein was significantly overexpressed in the epithelial section of T-OLP (*p* = 0.007 (Figure 4a,c), but not in the subepithelial compartment compared to N-OLP (*p* = 0.25) (Figure 4b). The expression difference was 9.78 times (Table 5). Additionally, this observed PD-L1 protein overexpression was significantly associated to malignant transformation within 5 years (Figure 4e). Results are summarized in Table 5.

High PD-1 expression was detected in the epithelial (*p*_E_ = 0.001) and subepithelial layers (*p*_S_ = 0.005) of T-OLP compared to N-OLP (Figure 3a,b). These results were confirmed by the AUC value of the ROC curve (AUC_E_ = 0.75, AUCs = 0.73) (Figure 3c). The overexpression in both tissue layers was significantly associated to malignant transformation (*p*_PD-1_E_ = 0.0001, *p*_PD-1_S_ = 0.02) (Figure 3d,e). Results are summarized in Table 5.

In order to examine the relationship between the expression of the checkpoints and DFS time of patients suffering from OLP was calculated. The mean disease free survival of the OLP patients was 77.2 weeks (range: 1–317 weeks). The group of T-OLP was divided in positive (overexpression) and negative (under expression) samples applying the calculated COP for each immune checkpoint. DFS time of the patients in regard to the expression of the different immune checkpoints was assessed (Table 6). Overexpression of PD-1 and PD-L1 in the epithelium resulted in a shortened DFS time of 40 weeks for high PD-L1 expression in the whole tissue (determined by RT qPCR), 47 weeks for epithelial overexpression of PD-L1 and 31 weeks for PD-1, respectively. Only overexpression of PD-1 in the subepithelial tissue of the OLP increased the DFS. However, the difference was only minimal at 7 weeks.

In order to determine the association of the expression of the different immune checkpoints Kaplan–Meier curves for disease-specific survival and a log-rank test was performed. No significant difference in survival was found between positive and negative T-OLP and PD-L1 mRNA expression (*p* = 0.35), PD-1 positive T-OLP neither in epithelial (*p* = 0.66) nor in the subepithelial (*p* = 0.51) section or expression of PD-L1 (*p* = 0.65) in the epithelium (Figure 5).

### 3.5. Association of Differential Expression Patterns with Histomorphological and Clinico-Histopathological Parameters

The overall expression of PD-L1 in the tissues examined by RT qPCR was not associated with the degree of dysplasia (grouped in high and low risk) in OLP (*p* = 0.521) and the differentiation (grading) (*p* = 0.751), the TNM classification or UICC of the OSCC (*p* > 0.05). However, there was a significant association of epithelial PD-L1 expression and the severity of dysplasia (*p*_E_ = 0.005). This could not be observed in the subepithelium (*p*_S_ = 0.762). UICC stadium (grouped) were related to the PD-L1 expression in OSCC (*p* = 0.028). Low epithelial PD-L1 levels were predominantly expressed in the early stage than in the late stage lesions. Subepithelial expression was not related (*p* = 0.647).

There was no significant correlation between OLP dysplasia (grouped) and PD-1 expression (*p*_E_ = 0.82; *p*_S_ = 0.831). Additionally, no correlation of PD-1 expression and TNM classification and grading could be seen (*p* > 0.05). The values were also examined for a possible relationship between UICC stage (early and late stage) and expression. With *p*_E_ = 0.616 and *p*_S_ = 0.0842 there was no significant association.

### 3.6. Inflammation in Relationship to Malignancy, Malignant Transformation and PD-1/PD-L1 Expression

#### 3.6.1. Association between Inflammation and Malignancy or Malignant Transformation

Based on the whole patients collective, distribution of the number of infiltrating inflammatory cells within the groups and its association to malignancy, occurrence of OLP and malignant transformation was evaluated. Out of the T-OLP 3/32 (9.4%) exhibited no, 11/32 (34.4%) exhibited mild, 14/11 (43.8%) exhibited moderate and 4/11 (12.5%) exhibited severe inflammation. Ten out of 49 (20.4%) out of the N-OLP were not, 23/49 (46.9%) were mildly, 13/49 (26.5%) were moderately and 3/49 (6.1%) were severely inflamed. Out of the OSCC all samples displayed inflammation (mild: 7/36 (19.4%), moderate: 23/36 (63.9%) and severe 6/36 (16.7%)), whereas NOM tissues presented only mild (16/35 (45.7%)) or no inflammation (19/35 (54.3%)). The χ square test revealed that the grade of infiltration with inflammatory cells was significantly increased in OLP and OSCC compared to NOM (*p* = 0.0001). The grade of inflammation was statistically associated to malignancy (*p* = 0.0001) and clinical manifestation of an OLP (NOM vs. T-OLP/N-OLP; s = 0.0001). There was no significant association between malignant transformation and grade of inflammation (T-OLP vs. N-OLP; *p*= 0.19). However, if OLP were grouped as low and high inflamed tissues the malignant transformation was significantly associated with inflammation (*p* = 0.035). Results are illustrated in Figure 6.

#### 3.6.2. Association between Inflammation and PD-L1 mRNA Expression

The expression of PD-L1 was analyzed by RT qPCR. Kruskal–Wallis test showed that there was a significant association between PD-L1 expression and inflammation (*p*= 0.0001). The expression rates rose with the increasing number of inflammatory cells (Figure 7). Significant differential PD-L1 expression rates were seen between all groups although only marginally between moderate and severe infiltrated tissues (*p* = 0.047). If OLP were grouped as low and high inflamed tissues expression was also significantly increased (Figure 7).

#### 3.6.3. Association between Inflammation and Protein Expression of PD-1 and PD-L1

If all groups are included in the evaluation, the Kruskal–Wallis test revealed that only the overexpression of PD-L1 in the epithelial part of the specimens is statistically associated with the severity of the inflammation (*p* = 0.0001). Epithelial expression was statistically correlated in all comparisons of inflammatory infiltrates except between no inflammation and mild inflammation (*p* = 0.071) (Figure 8). No statistically relevant association between the expression of PD-L1 in the subepithelium or PD-1 neither epithelial (*p* = 0.33) nor subepithelial (*p* = 0.07) could be demonstrated (data not shown). The same results were obtained after grouping the samples in high and low inflammation. Only PD-L1 was significantly overexpressed in the epithelial portion of highly inflamed tissues compared to not or mildly inflamed ones (*p* = 0.0001, Figure 8).

Other results were obtained if only the groups of OLP were included in the analysis. The comparison between the transforming and non-transforming group (T-OLP vs. N-OLP) revealed that 50% (24/48) of the T-OLPs were highly inflamed, whereas only 30.2% (16/53) of the N-OLPs did so. The association of malignant transformation and inflammation was significant (*p* = 0.04). Statistical analysis using the Kruskal–Wallis test showed a significant relationship between overexpression of PD-1 (epithelial *p* = 0.02 and subepithelial *p* = 0.01) (data not shown). These results could be confirmed by the MWU test if the cases were divided into lowly and highly inflamed OLP tissues (Figure 8). A significant relation of PD-L1 expression to infiltration has been proven in the epithelial compartment (*p* = 0.003) of the OLP but not for the subepithelial one (*p* = 0.576). After grouping into low and highly inflamed tissues, a significant association with the increased expression of PD-L1 in the epithelium except between none and mild inflammation was confirmed (Figure 8).

#### 3.6.4. Association between Localization of the OLP with Malignant Transformation and PD-1/PD-L1 Expression

Localizations of the OLP included in this part of investigations are listed in Table 1. For all these samples PCR and IHC expression analyses for PD-L1 were done. The highest rate of malignant transformation was shown for lesions of floor of the mouth (65.4%; 5/7), followed by those of the tongue (71.4%; 17/26), of the common oral cavity (33.3%; 4/12), the vestibule oris (25%; 1/4) and maxilla (25%; 2/8), the mandible (12.5%; 1/8) and buccal mucosa 11.8% (2/17). The association was significant between localization and transformation (*p* = 0.003) (Figure 9).

The Kruskal–Wallis test revealed that the expression rates of the immunomodulatory actors were not significantly changed in regard to the localization of the specific OLP (*p*_∆PD-L1_ = 0.32, P_PD-1_E_ = 0.37, P_PD-1_S_ = 0.75, P_PD-L1_E_ = 0.55, P_PD-L1_S_ = 0.37). If the different localizations were compared to each other, the MWU test showed no difference in the expression levels of the immunomodulators tested between the groups, neither in the PCR nor in the immunohistochemically investigation (data not shown).

## 4. Discussion

### 4.1. Implication of Altered Checkpoint Expression in OLP

The current analysis shows that OLP with malignant transformation (T-OLP) display a significantly increased PD-1 expression compared to N-OLP. This was observed in the epithelial and in the subepithelial compartment. Activated T cells show an upregulation of their PD-1 expression [30]. The PD-1 receptor is primarily expressed by T cells [30]. However, other inflammatory cells like B cells, natural killer (NK) cells and antigen presenting immune cells (APCs) can also express PD-1 [30]. A constantly high PD-1 expression is seen in T cells that are in a status of “T cell exhaustion”, which is associated with a decreased effector function and lacking antitumor immunity [31,32]. This indicates that the increased PD-1 expression in T-OLP observed in the current analysis is an expression of T cell exhaustion that might contribute to the malignant transformation of OLP. This applies to both epithelial and subepithelial PD-1-expressing T cells.

Previous data show that T-OLP has an increased macrophage density and a shift towards the immune-tolerant M2 polarization of macrophages [23]. These data indicate that T-OLP might have a compromised T cell and macrophage immunity, which could directly contribute to the process of malignant transformation.

With regard to immunosuppressive checkpoint ligands, an increased expression of PD-L1 was detected in the epithelial compartment of T-OLP compared to N-OLP in immunohistochemistry. In the subepithelial compartment, no significant differences in PD-L1 expression rates between both OLP groups were seen. Interestingly, the RT-PCR analysis revealed no significant difference in PD-L1 mRNA expression between T-OLP and N-OLP. This could be explained by the fact that epithelial and subepithelial tissue was included in the RT-PCR analyses. This underlines the need for distinguishing different tissue compartments (e.g., epithelial vs. subepithelial) when investigating immunologic parameters. This can be facilitated by a differentiating immunohistochemical approach. Additionally, OLP with a high degree of dysplasia also showed an increased degree of PD-L1 expression. This indicates, that an accumulation of genetic aberrations in oral epithelial cells that occurs during malignant transformation might be accompanied by an increase in PD-L1 expression.

PD-L1 is expressed by antigen presenting cells like macrophages but also by tumor cells [33]. In the current study, the epithelial cells were the dominant PD-L1 expressing cells in OLP. PD-L1 binds to the PD-1 receptor of activated T cells. This reduces the infiltration and the proliferation of effector T cells and thus contributes to the tumor immune-escape [33]. In OSCC, there is an association of PD-L1 expression with the infiltration of immune-tolerant “M2-like” polarized macrophages was shown [34]. Of note, previous analyses in OLP revealed an increased macrophage infiltration and M2 polarization in transforming OLP (T-OLP) [23]. Besides changes in PD1 and PD-L1 balance, differences in macrophage infiltration and macrophage polarization might also be involved in OSCC progression [10,11,35]. 

Cancers are caused by somatic mutations that can result in the expression of tumor-associated antigens that drive tumor progression and might also contribute to tumor initiation. These immunogenic neoantigens lead to specific T cell responses [36]. Additionally, it is demonstrated that the risk of cancer development is associated with a higher degree of inflammation of OLP [37]. Thus, at the same time when genetically altered cells are eliminated by the immune system, the immune response also increases inflammation, which leads to the destruction of tissue structures. To prevent critical damage, immunomodulatory proteins are expressed to induce suppression of the immune response, resulting in an immunosuppressive microenvironment in the OLP. As a consequence, the balance between elimination of tumor cells and malignant progenitor cells and their proliferation is disturbed. This disturbance in immunoediting could contribute to the escape of the altered cells from the immune system. As a consequence, malignant transformation occurs.

Lenouvel et al. found that increased PD-L1 expression was associated with the epithelium adjacent to tumor invasion. The presence of PD-L1 in the adjacent epithelium suggests that it may have been present before invasion and could have played a role in cancer development and progression. Hence, they suggest that the presence of the immune checkpoint protein in the adjacent non-tumor epithelium could be an indication of its participation in early carcinogenesis and metastasis [38]. PD-L1 expression has been analyzed in potentially malignant tissues of the oral cavity and, like in our study, an increased PD-L1 expression could be demonstrated in epithelial dysplasia compared to controls [26,39,40,41,42]. Additionally, we have been able to show that PD-L1 was significantly overexpressed in the epithelial compartment of transforming OLP compared to those that did not transform into malignancy within 5 years. In addition, its expression was significantly associated with malignant transformation. Hence, it could be postulated that PD-L1 overexpression in epithelial cells of OLP induces the local immunosuppression in the epithelium and triggers malignant transformation and early tumor development. Due to the hypothesis that immune evasion is considered as a hallmark of cancer, further research into immune checkpoint function in oral potentially malignant lesions is justified.

### 4.2. Predictive Value of Checkpoints in OLP

By determining a cut-off point (COP) for PD-L1 and PD-1 expression and allocating the individual cases to the positive (increased expression) and negative group, the value of both parameters as a predictive marker for malignant transformation of individual OLP cases could be evaluated. The χ2-test revealed that PD-L1 and PD-1 could serve as predictive markers for OLP malignant transformation with acceptable sensitivity and specificity. Additionally, DFS time of patients of the T-OLP group who were positive for epithelial overexpression of PD-1 and PD-L1 was shortened. However, Kaplan–Meier graphs and log-rank-tests revealed no significant association between DFS time and positivity for immune checkpoint expression. This could already been shown for tumors, where the PD-L1 staining was not significantly associated with DFS [38]. Hence, the overexpression of the immune modulators may have no effect on the upset of the disease. However, one can suggest that the immunosuppression induced by expression of the checkpoints leads to an immediate, rapid expansion of the malignant cells already present in the OLP and that the tumor shows itself clinically in a timely manner.

Therefore, PD-L1 and PD-1 should be evaluated as predictive parameters in prospective analyses. However, it needs to be considered that other parameters like the melanoma associated antigen A (MAGE-A) expression revealed higher sensitivity and specificity in retrospective studies [43,44]. 

### 4.3. Therapeutic Potential

There is currently no evidence that any treatment can counteract the malignant transformation of OLP and prevent the development of oral cavity carcinomas [45]. Therefore, our study additionally aims to enable new effective therapies based on immunological features. The fact that T-OLP and N-OLP differ regarding PD-1-, PD-L1- expression and macrophage polarization [23] indicates that immunologic alterations precede the malignant transformation of OLP. This could open a window for immunotherapy in OSCC precursor lesions. In non-muscle invasive bladder cancer local immunotherapy using Bacillus Calmette–Guérin (BCG) is an established treatment option [46]. A combination of the toll like receptor (TLR) activating BCG with blocking of PD-L1/PD-1 signaling might additionally increase the efficiency of the treatment [46]. However, there are conflicting data regarding the association of cancer PD-L1 expression with response to classical BCG treatment [47,48]. Recent data indicate that colocalization of PD-L1 with CD8 might be an indicator for resistance against BCG therapy [48]. 

Inhibitory checkpoint molecules are promising targets for cancer immunotherapy. The clinical application of immune-checkpoint inhibitors has also dramatically improved the treatment of patients suffering from OSCC. In this regard the most important immune checkpoint proteins identified are CTLA-4 and the members of the PD-1/PD-L1 axis. In the current study, an enhanced expression of PD-L1 was seen T-OLP and OSCC. Moreover, it was shown that this checkpoint was expressed even higher in T-OLP than in OSCC. Therefore, it could be suggested that the treatment applying anti-immune checkpoint antibodies should be equally effective in treating OLP, as it works to counteract the immunosuppression in the OLP microenvironment and would allow the immune system to get rid of the altered dysplastic cells. This would stop the progression and may even lead to regression of the lesion [49]. The anti-PD/PD-L immunotherapy could also be used in treating multifocal lesions and large lesions, where the complete surgical removal is not possible due to anatomical structures. Additionally, undetected altered cells would be targeted and removed by the immune system and prevent field cancerization. However, in order to establish this as an effective treatment for OLP and to determine the possible side effects, prospective clinical studies with a long follow-up period are urgently needed.

### 4.4. Association between Inflammation and Checkpoint Expression

An increased degree of inflammation has been associated with the severity of oral lesions. The immunosuppression induced by chronic inflammation was thought to play a role in the progression of OLP to OSCC [37]. The fact that the PD/PD-L signaling pathway works similarly by creating a tumor microenvironment of immune suppression makes it very plausible that there is a positive association between the expression of PD-L1, inflammation and malignant transformation [37,50,51]. 

The results of this study also support the hypothesis that a higher inflammation rate is related to malignant potential, as most of the OSCC and more than half of the T-OLP have moderate to severe inflammation. The association between the malignant transformation and the degree of inflammation was statistically significant. Moreover, a statistically significant association was found between the expression of PD-L1/PD-1 and the degree of inflammation. These findings are in accordance with previous studies that postulate the importance of inflammation as a factor in carcinogenesis [52]. In summary, it could be postulated that the expression of PD-L1/PD-1 and the expression of inflammation markers are closely intertwined and thus should be examined in relation to one another in future research to better understand their interactions and how they influence each other. Additionally, it could be argued that the relationship between inflammation and PD-L1/PD-1 expression supports the effectiveness of anti-PD/PD-L treatment, as it has been established that both a higher degree of inflammation and elevated PD-L1/PD-1 levels were prevalent in both OSCC and T-OLP.

### 4.5. Association between Localization of the OLP, Malignant Transformation and Immune Checkpoint Expression

It is well established that the risk of malignant transformation is significantly associated to the localization of OLP. Thus, OLP, which developed on the tongue or at the floor of mouth progress more often into malignancy as OLP at other sides of the oral cavity [53]. Additionally, positive PD-L1 staining was significantly more likely in tongue squamous cell carcinomas [38]. This could also be shown in this study. OLP of the floor of the mouth had the highest rates of malignant progression followed by those of the tongue. This association was additionally significant. However, no significant association of the localization and altered expression of PD-1 and PD-L1 neither by RT qPCR or IHC could be detected. To the best of the authors’ knowledge, this study is the first to investigate this relationship. The observations suggest that increased expression of checkpoints is involved in or at least promotes malignant transformation, but that this mechanism is independent of the localization of the OLP and thus represents a general immunological event. One reason for this could be that the immunological mechanism is independent of acting mutagenic noxious agents, which are supposed to be responsible for the increased transformation rates at these localizations.

### 4.6. Limitations of the Study

Smoking and alcohol abuse drastically increase the risk of malignant transformation. Hence, these habits have to be taken into account when assessing the risk of malignant transformation. These data were not collected in this study. This could be seen as a shortcoming of the investigation. However, the present study was only intended to show the influence of the PD1/PD-L1 axis as an immunosuppressive agitator on malignant transformation and whether the dignity of a potentially malignant lesion can be better assessed by detecting overexpression of these genes. This question was investigated independently of consumer behavior. Nevertheless, these parameters will be considered in our further multicenter study.

Due to the limited material, it was not possible to perform expression analyses for all checkpoints in each sample. For this reason, the cases included in the expression analyses vary greatly. This could reduce the significance of the results. In our planned prospective study, this pitfall will no longer occur, as the material will be used exclusively for the expression studies with the selected markers and all samples taken from patients will not only be fixed and embedded in formalin, but also archived as tissue samples in RNA later.

Markers of T and MDSC cells may help to elucidate the status of immune cells. However, we did not perform any expression analysis for CD8 until now. However, we are going to carry out a study, which aims to investigate the expression of several T cell (e.g., CD8, CD4 and CD3) and MDSC markers (e.g., CD33 and CD115) in T-OLP and N-OLP.

All over, the number of samples included in the analyses limited our findings particularly for survival. Further studies are needed to confirm the results. However, we are confident we reached a sufficient sample size in our started predictive study to clarify the prognostic impact of checkpoint expression.

## 5. Conclusions

The data indicated that upregulation of PD-L1 may be associated with disease progress in oral potentially malignant disorders and malignant lesions. A high degree of inflammation in OLP was associated with high expression of PD-L1, PD-1 and malignant transformation. A differential immunohistochemical approach is helpful for the analysis of immune cells in cancer precursor lesions.

PD-1/PD-L1 may represent a prognostic indicator to determine the risk of malignant progression of OLP. Local immunosuppression in the epithelium could be induced by PD-L1 overexpression in epithelial cells and trigger malignant transformation. Hence, checkpoint inhibitors could counteract tumor development and serve as therapeutic agents in patients with high-risk OLP lesions.

## Figures and Tables

**Figure 1 biomedicines-09-00194-f001:**
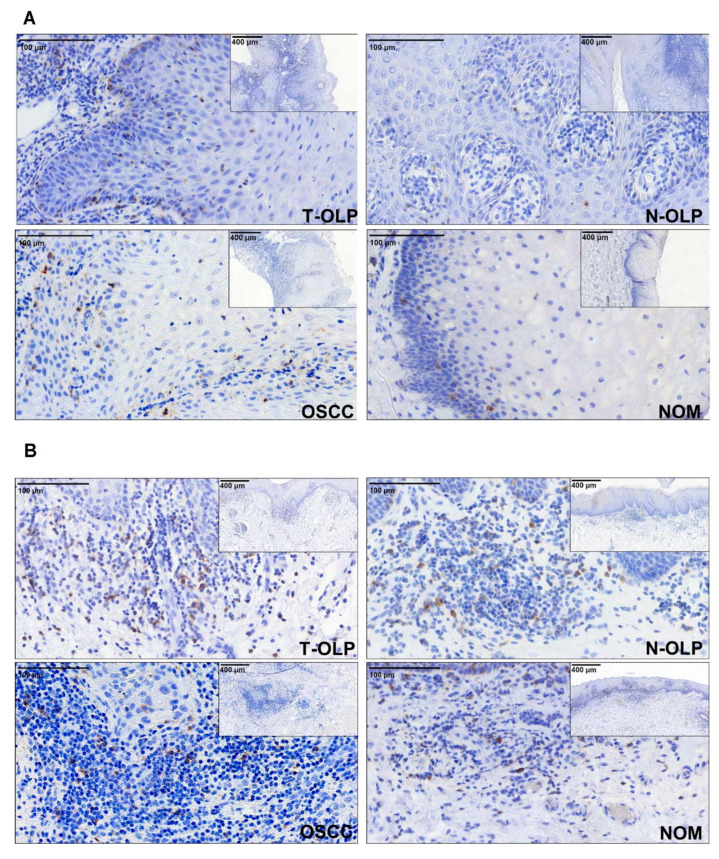
Immunohistochemical expression analysis of PD-1. The expression of the proteins was separately analyzed in the epithelium (**A**) and the subepithelium (**B**). Differential expression between the different tissues was observed.

**Figure 2 biomedicines-09-00194-f002:**
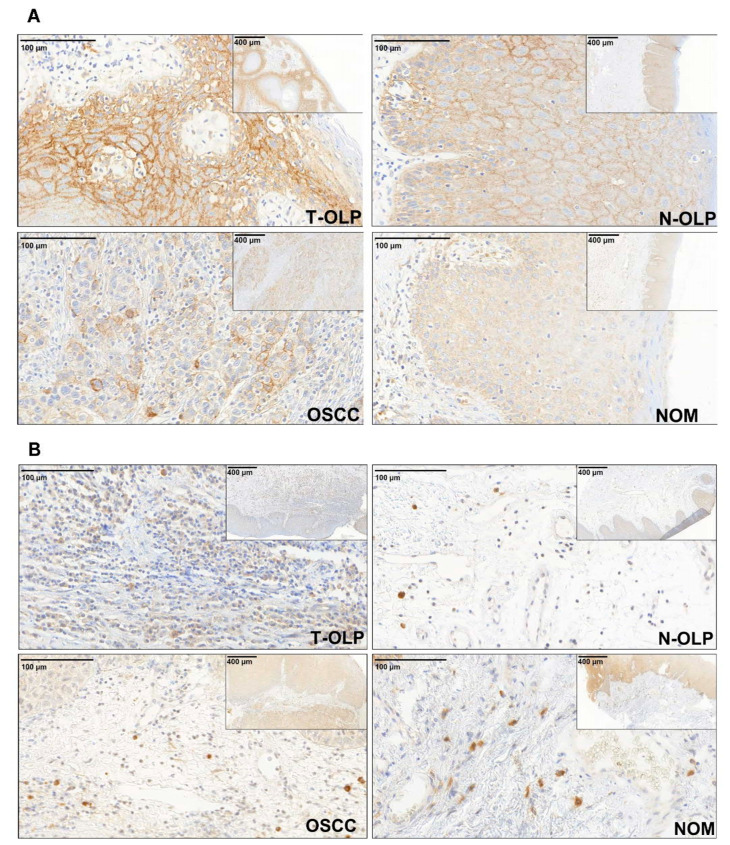
Immunohistochemical expression analysis of PD-L1. The expression of the protein was separately analyzed in the epithelial (**A**) and subepithelial (**B**) compartment. The expression varies in the different tissue specimens. PD-L1 is visibly overexpressed in the epithelial compartment of T-OLP and OSCC compared to N-OLP and NOM.

**Figure 3 biomedicines-09-00194-f003:**
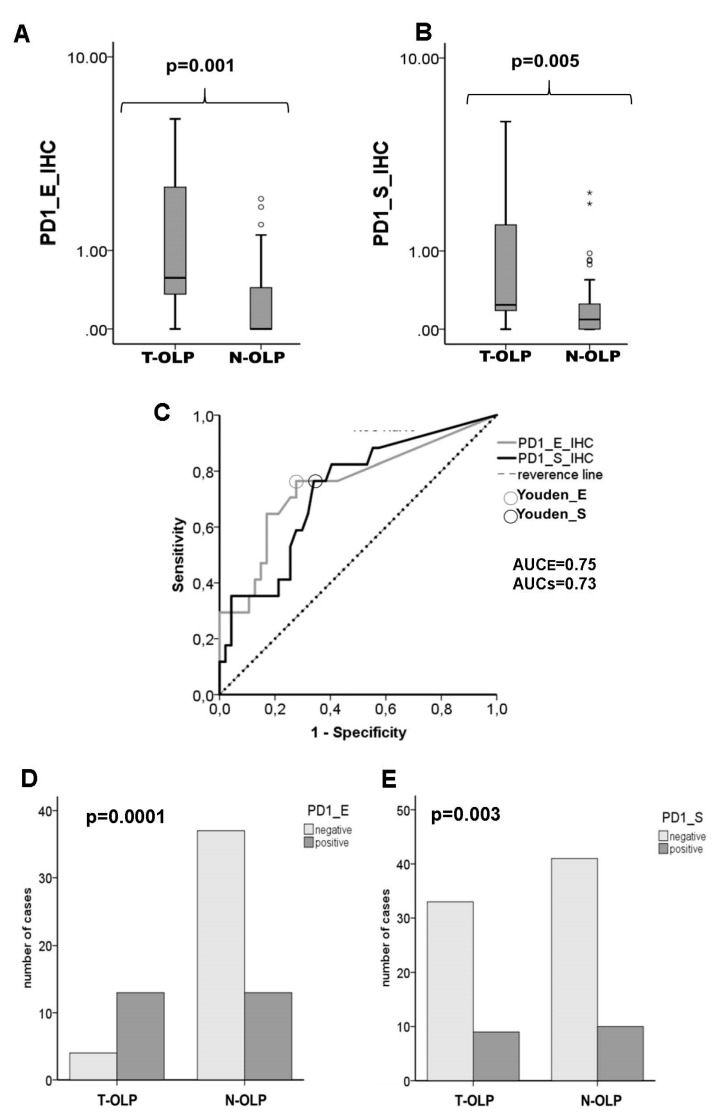
Comparison of the expression rates of PD-1 in the epithelium and the subepithelium of T-OLP and N-OLP determined by IHC. Boxplots and MWU-tests show a significant epithelial (**A**) and subepithelial (**B**) overexpression of PD-1 in the T-OLP group compared to N-OLP. (**C**) Receiver operating characteristic (ROC) curve and AUC value indicate a significant relation between PD-1 protein expression and malignant transformation in epithelial and subepithelial compartment (T-OLP (positive status) vs. N-OLP). A significant association between malignant transformation and protein overexpression in the epithelial (**D**) and subepithelial sections (**E**) was seen (χ2-test).

**Figure 4 biomedicines-09-00194-f004:**
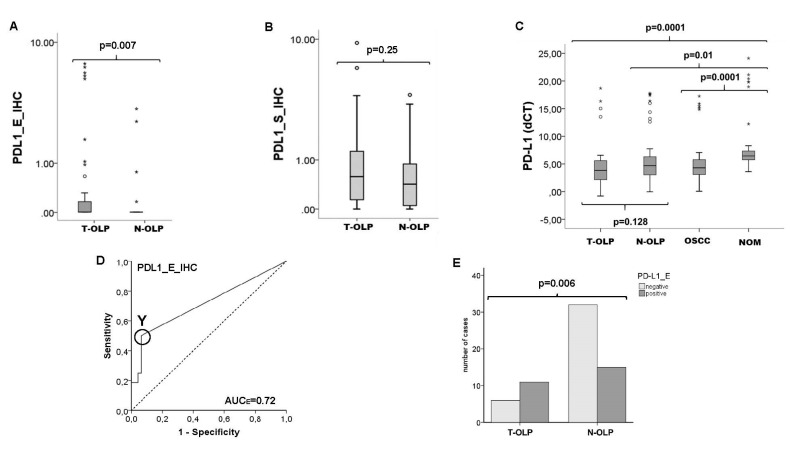
Comparison of the expression rates of PD-L1 in the epithelium and the subepithelium of T-OLP and N-OLP. Comparison of the protein expression rates of PD-L1 in the epithelium (**A**) and the subepithelium (**B**) of T-OLP and N-OLP PD-L1 was significantly overexpressed in epithelial compartment. (**C**) RT qPCR expression analysis in all tissue samples of the groups provide no significant difference in the expression level in T-OLP compared to N-OLP. (**D**) ROC, AUC value confirms the significant association between epithelial protein overexpression and malignant trans-formation (T-OLP (positive status) vs. N-OLP). (**E**) Malignant transformation is significantly associated with epithelial overexpression of PD-L1 protein (χ2-test). o stands for an outlier; * indicates an extreme value.

**Figure 5 biomedicines-09-00194-f005:**
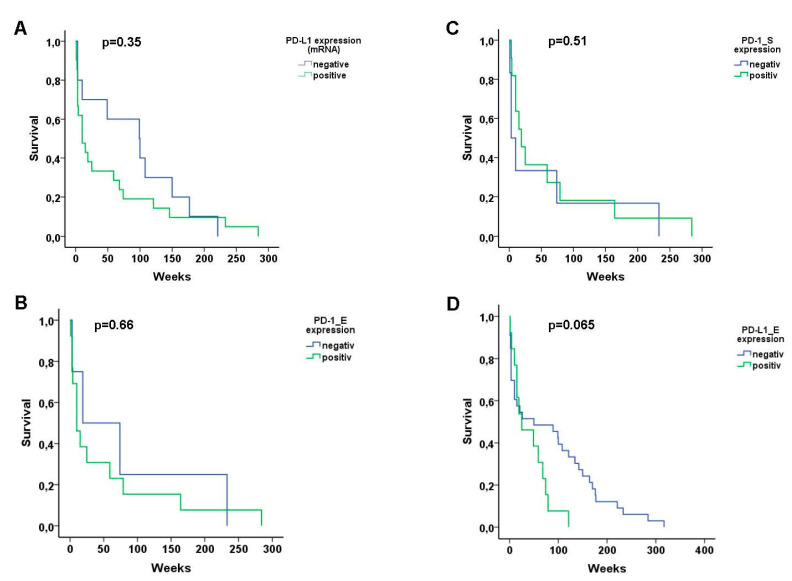
Kaplan–Meier curves for disease-specific survival. No significant difference in survival of the patients was found between (**A**) PD-L1 mRNA positive and negative T-OLP, PD-1 positive or negative T-OLP neither (**B**) in epithelial nor (**C**) in the subepithelial section or (**D**) positive and negative expression of PD-L1 in the epithelia, respectively.

**Figure 6 biomedicines-09-00194-f006:**
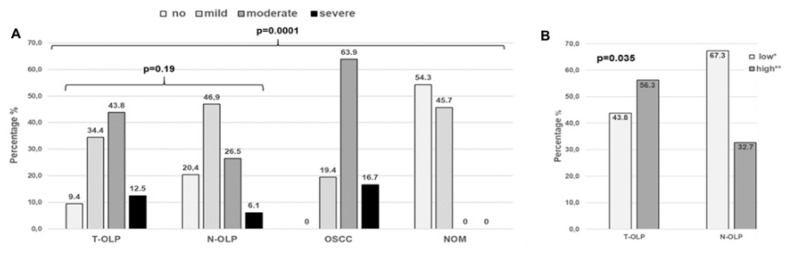
Association between inflammation and malignant transformation or malignancy, respectively. There was a statistical significant association between malignancy but not between malignant transformation and inflammatory infiltration over all groups of inflammatory degrees (**A**). If OLP were grouped into high and low infiltration by inflammatory cells a significant association between inflammation and malignant transformation could be shown. (**B**). * Group of no and mild inflamed; ** high inflammation (moderate/high).

**Figure 7 biomedicines-09-00194-f007:**
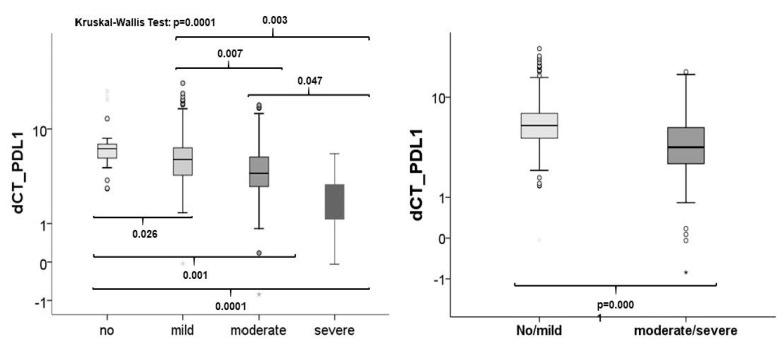
Association between inflammation and checkpoint expression. The comparison of the expression rates of all tissue sample groups to each other revealed that PD-L1 is statistically differentially expressed in tissues with diverse grades of inflammation (**left**). If OLP were grouped into low (no/mild) and high infiltration (moderate/severe) by inflammatory cells, a significant association between inflammation and expression could be seen (**right**).

**Figure 8 biomedicines-09-00194-f008:**
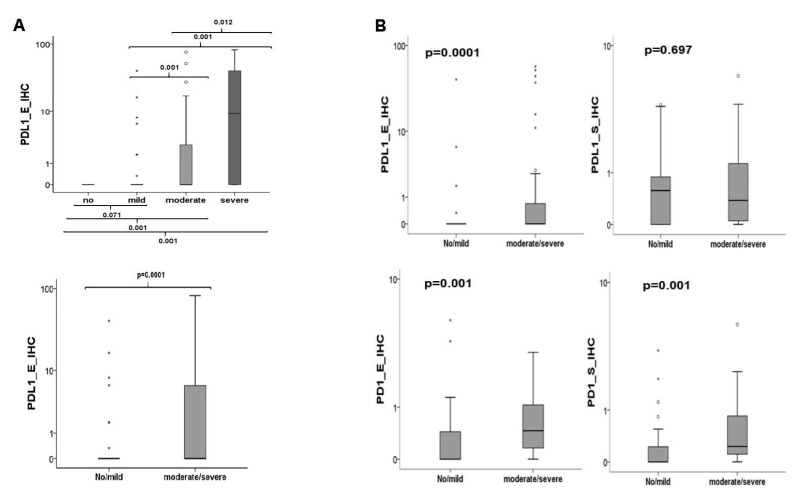
Association between inflammation and checkpoint expression. (**A**) If all groups are included in the evaluation PD-L1 expression in the epithelial part of the specimens is statistically associated with the severity of inflammation. (**B**) If only the groups of OLP were included in the analysis a significant relationship between overexpression of PD-1 (epithelial/subepithelial) and of PD-L1 (epithelial) to high inflammation could be seen.

**Figure 9 biomedicines-09-00194-f009:**
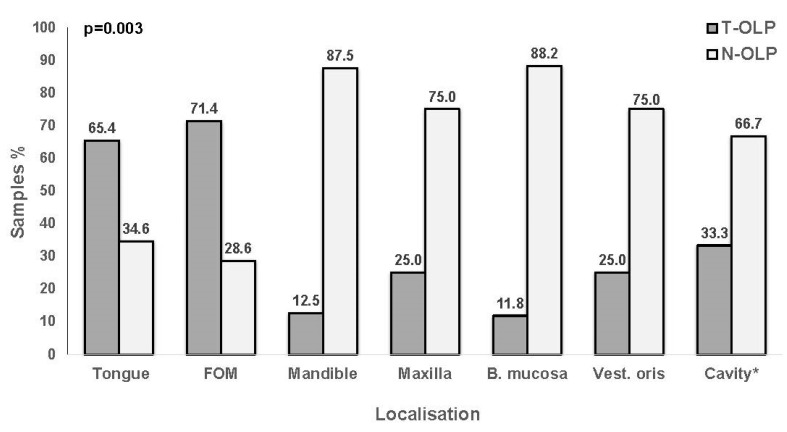
Relationship between localization and malignant transformation. OLP localized to the tongue and floor of the mouth (FOM) proceed into malignancy more frequently than those occurring in the mandible, maxilla, buccal mucosa (B. mucosa), vestibulum oris (Vest. oris) or other parts of the *oral cavity. Malignant transformation is significantly associated with the localization of the lesion (χ2-test).

**Table 1 biomedicines-09-00194-t001:** Demographic, clinical and histopathological characteristics of the histological patients’ collective used in RT qPCR.

Group		T-OLP	N-OLP	OSCC	NOM	Total
**Number of Cases**		32	50	38	42	162
**Sex**	Male	20	24	24	-	-
	Female	12	14	18	-	-
**Collective**	Erlangen	17	21	42	-	-
	Halle	15	17	0	-	-
**Mean Age/SD**		62/10.9	55/12.4	66/11.2	43/20.4	-
**Age Range**		45–92	23–81	34–93	18–81	-
**Localization**	Tongue	17	9	-	-	-
	Floor of mouth	5	2	-	-	-
	Lower jaw	1	7	-	-	-
	Upper jaw	2	6	-	-	-
	Buccal mucosa	2	15	-	-	-
	Vestibulum oris	1	3	-	-	-
	Oral cavity *	4	8	-	-	-
**Dysplasia**	D0	17	37	-	-	-
	D1	6	11	-	-	-
	D2	4	2	-	-	-

All these samples were included in PD-L1 expression analysis by immunohistochemistry (IHC). General statistical analyses concerning localization and inflammation and the association of these parameters to malignant transformation, and disease free survival (DFS) (time until malignant transformation), Kaplan–Meier graph and log rank test for survival were carried out with the smaller PCR collective. IF = inflammation, inflammatory infiltration (grade 0–3), * the localization was not further determined.

**Table 2 biomedicines-09-00194-t002:** Demographic, clinical and histopathological characteristics of the histological patients’ collective used in immunohistochemistry.

Group		T-OLP		N-OLP		OSCC		NOM		Total	
		PD-1	PD-L1	PD-1	PD-L1	PD-1	PD-L1	PD-1	PD-L1	PD-1	PD-L1
**Number of Cases ***		17	46	50	53	21	45	20	20	108	164
**Sex**	Male	9	29	25	27	11	28	10	10	55	94
	Female	8	17	25	26	10	17	10	10	53	70
**Collective**	Erlangen	5	27	30	31	9	26	20	20	64	104
	Halle	12	19	20	22	12	19	0	0	44	60
**Mean Age/SD**		56.5/15.1		53.7/12.6		60.9/14.8		43.1/19.8		-	-
**Age Range**		45–93		23–81		38–93		18–81		-	-
		**PD-1**	**PD-L1**	**PD-1**	**PD-L1**	**PD-1**	**PD-L1**	**PD-1**	**PD-L1**		
**Dysplasia**	D0	10	27	36	39	-		-		-	-
	D1	2	8	13	13					-	-
	D2	2	6	1	1					-	-
	D3	3	5	0	0					-	-
										-	-
**High Risk ***	D0/D1	12	35	49	52					-	-
**Low Risk ***	D2/D3	5	11	1	1					-	-
**Grading**	G1	-				7	13	-		-	-
	G2					8	18			-	-
	G3					5	11			-	-
	n.d.					1	3			-	-
**T-Status ***	T1-T2	-				18		-		-	-
	T3-T4					2				-	-
	n.d.					1	8			-	-
**N-Status ***	N0	-				10	18	-		-	-
	N+					11	4			-	-
	UK					-	23			-	-
**Inflammation**	N-IF	2	6	10	11	0	0	9	9	-	-
	M-IF	5	17	24	25	2	7	11	11	-	-
	Mo-IF	7	16	13	13	14	31	0	0	-	-
	S-IF	3	5	3	3	5	7	0	0	-	-
	n.d.		2		1					-	-
**Clinical Stage ***	Early	-				17	30	-		-	-
	Late					3	8			-	-
	n.d.					1	7			-	-

IF = inflammation, inflammatory infiltration (grade 0–3), N = no, M = mild, Mo = moderate, S = severe; n.d. = unknown; D0/D1 = Low risk; D2/D3 = High risk of transformation; * grouped. Not all samples examined for PD-L1 protein expression were investigated by PCR.

**Table 3 biomedicines-09-00194-t003:** Descriptive statistics for average expression rates of ∆LI and ∆CT_PD-L1_ values for all groups determined by immunohistochemistry and RT-qPCR.

Group		PD1 (ΔLI)		PD-L1 (ΔLI)		PD-L1 (∆CT)
		Epithelial	Subepithelial	Epithelial	Subepithelial	Whole Tissue
T-OLP	Mean	1.32	1.01	6.07	1.27	4.86
	SD	1.58	1.46	15.58	1.91	4.67
	N	17	17	46	46	32
N-OLP	Mean	0.3	0.26	0.62	0.69	6.47
	SD	0.52	0.49	2.81	0.92	5.29
	N	50	50	53	53	50
OSCC	Mean	0.37	0.25	10.38	0.47	5.76
	SD	0.8	0.38	19.63	0.67	4.4
	N	21	21	45	45	38
NOM	Mean	1	0.66	0.1	1.59	8.37
	SD	1.21	1.1	0.42	1.73	5.32
	N	20	20	20	20	24

In IHC were separately investigated in the epithelial and subepithelial compartment, *N* = number of cases. ΔLI = average labeling index, ∆CT = average cycle threshold, SD = standard deviation.

**Table 4 biomedicines-09-00194-t004:** Comparison of the expression rate of PD-L1 and PD-1 between NOM and the OSCC, T-OLP and N-OLP group.

vs. NOM	*p*-Value MWU	FC Up/Down	AUC	*p*-Value (χ2)
**RT qPCR (PD-L1)**				
**T-OLP**	0.0001	11.4	0.70	0.0001
**N-OLP**	0.001	3.8	0.65	0.01
**OSCC**	0.0001	8.9	0.77	0.0001
**IHC (PD-L1 epithelial)**				
**T-OLP**	0.04	60.14	0.617	0.03
**N-OLP**	0.703	6.27	n.d	n.d
**OSCC**	0.002	99.41	0.703	0.003
**IHC (PD-L1 subepithelial)**				
**T-OLP**	0.19	0.8/−1.3	n.d	n.d
**N-OLP**	0.014	0.44/−2.3	0.295	n.d.
**OSCC**	0.002	0.29/−3.4	0.233	n.d.
**IHC (PD-1 epithelial)**				
**T-OLP**	0.729	1.27	n.d	n.d
**N-OLP**	0.0001	0.29/−3.5	0.203	n.d.
**OSCC**	0.002	0.36/−2.8	0.194	n.d.
**IHC (PD-1 subepithelial)**				
**T-OLP**	0.55	1.52	n.d	n.d
**N-OLP**	0.014	0.40/−2.6	0.309	n.d.
**OSCC**	0.074	0.37/−2.8	n.d	n.d

E = epithelium; S = subepithelium, FC = fold change (negative values mean downregulation in comparison to NOM).

**Table 5 biomedicines-09-00194-t005:** Statistical results of the comparison of PD-1 and PD-L1 Expression between NOM and T-OLP, N-OLP and OSCC.

Target	*p*-Value MWU	FC	AUC	Sensitivity	Specificity	COP	*p*-Value (χ2)
T-OLP vs. N-OLP							
PD-1_E (IHC)	0.001	4.41	0.75	76.5	72.3	0.34	0.0001
PD-1_S (IHC)	0.005	3.83	0.78	76.5	66	0.18	0.002
PD-L1 (qPCR)	0.128	3.05	n.d.	n.d.	n.d.	n.d.	n.d.
PD-L1_E (IHC)	0.006	9.73	0.72	50.0	93.6	0.16	0.006
PD-L1_S (IHC)	0.25	1.83	n.d.	n.d.	n.d.	n.d.	n.d.

E = epithelium; S = subepithelium, FC = fold change, COP = Cut of Point. AUC = Area under the curve.

**Table 6 biomedicines-09-00194-t006:** Disease free survival of the patients in regard to overexpression or the different immune checkpoints.

DFS of Patients in Weeks (Mean 77.2)			
Checkpoint	Expression	Range	Mean DFS
PD-L1-PCR			
* COP = 5.027	+	1–284	52
	−	3–221	92
PD-1_E			
** COP = 0.34	+	1–284	51.3
	−	3–233	82.3
PD-1_S			
** COP = 0.18	+	3–284	61.1
	−	1–233	54
PD-L1_E			
** COP = 0.16	+	1–121	39.1
	−	1–317	86.3

The group of T-OLP were divided in positive and negative samples applying the calculated COP. * COP was calculated via the comparison OLP against NOM (∆CT = 5.027); ** COP (LI) was calculated via the comparison N-OLP against T-OLP. E = epithelia; S = subepithelia.

## Data Availability

Additional data will be provided by the corresponding author on request.

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
