# Peer review of "Importance of the PD-1/PD-L1 Axis for Malignant Transformation and Risk Assessment of Oral Leukoplakia"

_biomedicines, 2021, doi:10.3390/biomedicines9020194_

Round 1

Reviewer 1 Report

This is a valuable study on the prognostic markers of oral leukoplakia.

  1. The overall presentation needs improvement
  2. The term "premalignant" should be replaced with "potentially malignant" in line with the WHO nomenclature.
  3. Aims of the study (p4) should be reworded. The second sentence under aims does not read as an aim of the study.
  4. The study spans 18 years in two hospitals. The authors do not state whether these biopsies were from a consecutive sample or a convenient sample taken from that long span of years. It is not clear whether there is any bias in sample selection. Some clarification on sample selection is needed.
  5. Was dysplasia grading transcribed from the original pathology notes or part of the research? If the tissues were re-graded you must indicate how many pathologists graded the tissues. It is recommended that the consensus between two pathologists is necessary for dysplasia grading. 
  6.   In Table 1 there are significant discrepancies between the number of cases included in the study and the numbers analyzed. For eg there were 38 OSCCs entered into the study but PDL1 expression was analyzed in 45 samples. It is necessary to explain these differences.
  7. Did the authors carry out any pilot studies to optimize antigen retrieval and antibody dilutions for IHC studies? These steps should be explained.
  8. IHC studies were done in two different centers. Was quality assurance checked by repeating IHC on the same samples in the two labs? Please explain.
  9.  In p9 the authors state "for both antigens membranous staining was defined as a positive result". Yet, Fig 1 supplied for PD1 does not illustrate any membranous staining. IHC staining shown is clearly nuclear. 
  10. The description of the results could be improved by not repeating what is included in the Tables.
  11. The results in Table 3 should not be expressed in 4 decimal points. The measurements would not have been accurate for more than one decimal point. 
  12. All data presented are univariate. The study lacks a multivariate analysis. For example smoking status of the 2 groups (T-OLP and N-OLP) is an important variable that has not been studied or discussed with reference to the expression of biomarkers or outcomes.
  13.  One does not know whether the T cells expressing these markers are alive or non-functional. This should be explained in the discussion. What other biomarkers may have helped to elucidate the functional status of T cells?
  14. Any association between current data and data from ref 17 should be better explained.
  15. Calculating epithelial LIs based on counting 100 stained epithelial cells indicates to me that data could not be representative. Could you give a reference to other studies that limited counting to 100 epithelial cells per section/case? Kindly discuss this limitation.
  16. The recent study on PDL1 on OSCC by Daniel Lenouvel (2020) describing their findings on margins of OSCC should be referenced and discussed.
  17. Reference to recent systematic reviews on biomarkers on malignant transformation of oral leukoplakia is lacking in the discussion.
  18. I am not impressed by the photomicrographs shown in Figs 1 and 2. Tangential sectioning is noted, and this would have impacted negatively on cell counts (LIs). If these are your very best sections, automated counting could be at risk.
  19. Minor: in p9 the terms used, epithelially and sub epithelially are grammatically incorrect. p5 line 109 the reference is incomplete. Should be Head and Neck.
  20. Any limitations of your study should be presented.  

Reviewer 2 Report

The manuscript entitled “Importance of the PD-1/PD-L1 axis for malignant transformation and risk assessment of oral leukoplakia” which was contributed Ries et al. describes the correlation between the expression of PD-1 and PD-L1 and malignant transformation in OLP.There are several issues in the methodology and in the presentation of the results that need to be addressed.

Comments:

  1. 1. I strongly encourage the authors to describe the lesion sites and habits of smoking and alcohol intake. Previous reports showed that lesions in the tongue and oral floor have a high risk of malignant transformation. Moreover, Smokers with OLP have an increased rate of malignant transformation in relation to nonsmokers.
  2. There should be a description how leukoplakia cases were managed. Any intervention (treatment) would affect the malignant transformation rate.
  3. Considering follow-up duration, authors should add Kaplan–Meier graph and log-rank test to examine the relationship between each gene expression and malignant transformation of OLP.
  4. If authors have an another data such as M2-macrophages markers and CD8, please add these data in this manuscript. It is more novel and understandable.
  5. In Materials and Methods section, the authors state that “Lastly, samples were histologically classified according to the inflammatory infiltration. The four categories were no, mild, moderate, and severe inflammation”. This classification is unclear and should be defined in detail.
  6. Because Figure 3 and 5 are unclear and small, the authors should present more clear images.
  7. In Table 4, the AUC of T-OLP vs NOM is “7.70”. The authors should correct it because AUC value lies between 0 to 1.
  8. English needs correction in several places. For instance, “und (lines 64 and 67)”, “PD-L (lines 270 and 272)”.

Round 2

Reviewer 2 Report

I appreciate the authors for their collaboration. However, in my opinion, it still needs some revisions to be acceptable for publication.

Comments:

  1. In Materials and Methods section, the authors state “Samples that showed inflammation were categorized into mild inflammation (number of inflammatory cells ≤ 10), moderate inflammation (inflammatory cells ≤ 30), and severe inflammation (inflammatory cells > 30)”. The authors should present representative images of each grade of inflammation and additionally describe how the number of inflammatory cells were counted. (magnification, etc)
  2. Please change “buccal cheek” to “buccal mucosa”.
  3. In Table4, please change “PD1” to “PD-1”.
